# Prognostic Relevance of Type 2 Diabetes and Metformin Treatment in Head and Neck Melanoma: Results from a Population-Based Cohort Study

Steffen Spoerl [1], Michael Gerken [2], Susanne Schimnitz [1], Juergen Taxis [1], René Fischer [3], Sophia R. Lindner [1], Tobias Ettl [1], Nils Ludwig [1], Silvia Spoerl [4], Torsten E. Reichert [1] and Gerrit Spanier [1,*]

[1] Department of Cranio-Maxillofacial Surgery, University Hospital Regensburg, 93053 Regensburg, Germany
[2] Tumor Center—Institute for Quality Management and Health Services Research, University of Regensburg, 93053 Regensburg, Germany
[3] Department of Otorhinolaryngology, University Hospital Regensburg, 93053 Regensburg, Germany
[4] Department of Internal Medicine 5—Hematology/Oncology, Friedrich-Alexander University Erlangen-Nürnberg, 91054 Erlangen, Germany
[*] Correspondence: gerrit.spanier@ukr.de

**Abstract:** Background: Type 2 Diabetes (DM2) and the consecutively daily use of antidiabetic medication are characterized by a frequent prevalence worldwide and were shown to impact the initiation and progression of malignant diseases. While these effects were observed in a variety of malignancies, comprehensive data about the role of DM2 and antidiabetic drugs in the outcome of head and neck melanoma (HNM) patients are missing. Methods: This retrospective population-based cohort study included 382 HNM patients from Eastern Bavaria having received tumor resection to negative margins between 2010 and 2017. Recurrence-free survival (RFS) was evaluated with regard to DM2 and routine metformin intake. Statistical analysis was performed by uni- and multivariate analyses. The median follow-up time was 5.6 years. Results: DM2 was diagnosed in 68 patients (17.8%), routine metformin intake was found in 39 cases (10.2%). The univariate survival analysis revealed impaired 5-year RFS in HNM patients with DM2 compared to non-diabetic controls ($p = 0.016$; 64.0% and 74.5%, respectively). The multivariate Cox regression substantiated this effect (HR = 1.980, 95% CI = 1.108–3.538, $p = 0.021$). In detail, the cumulative locoregional recurrence rate displayed the most far-reaching negative effect on the RFS of diabetic HNM patients (HR = 4.173, 95% CI = 1.628–10.697, $p = 0.003$). For metformin intake, a profound positive effect on the RFS in multivariate statistics was observed, both in the complete cohort (HR = 0.396, 95% CI = 0.177–0.884, $p = 0.024$) as well as in the cohort of diabetic HNM patients (HR = 0.352, 95% CI = 0.135–0.913, $p = 0.032$). Conclusions: This study emphasizes that DM2 is a relevant comorbid condition in HNM patients, impairing patient survival. Metformin intake was associated with a favorable outcome in HNM patients, providing possible therapeutic implications for future adjuvant treatment regimes.

**Keywords:** head and neck melanoma; type 2 diabetes; metformin; outcome; survival; recurrence

## 1. Introduction

Due to its worldwide prevalence of around 400 million people, diabetes represents a major challenge of health care in the 21st century [1]. Especially, Type 2 Diabetes (DM2) accounts for a broad majority of cases with glucose disorders and, characterized as a civilization disease, is considered to be closely related to a wide variety of complications including ophthalmological, nephrological, and neurological disorders. Predominantly, macrovascular consequences, resulting in severe cardiovascular diseases, account for one of the most frequent causes of acute hospitalization in health care, not only in developed countries [1]. Analogous to DM2, malignancies are among the most frequent diseases worldwide and, interestingly, the recent literature indicates a possible link between DM2

and cancer. Remarkable efforts in clinical and translational research identified DM2 as an independent risk factor for developing a variety of malignant disorders: for pancreatic and liver cancer, large cohort studies and meta analyses hereby attributed to diabetic patients a profound risk to develop malignant neoplasms [2–4]. In this regard, various pathways had been analyzed; however, the interaction of elevated blood glucose, insulin resistance, and obesity might play a crucial role for the initiation of carcinogenesis [5,6]. Aside of the increased chances for developing malignancies, the role of DM2 in the outcome of cancer patients is, especially because of the increasing prevalence of DM2 in an aging population, of particular interest [7]. Hereby, numerous studies attributed to patients with diabetic conditions a less favorable outcome compared to non-diabetic controls [8,9]. However, especially in determining the outcome of cancer patients with DM2, the literature is quite inconsistent. For breast as well as colorectal cancer, retrospective studies attributed no significant deleterious effects to a deranged glucose metabolism [10,11].

Besides the sheer presence of DM2 affecting the outcome of cancer patients, emerging interest in evaluating the effects of antidiabetic drugs in cancer patients, exists. The drug of choice in this regard is the frequently used biguanide metformin [12]. Since the first reports about the possible antineoplastic effects of metformin arose in the early 2000s [13–15], extensive research has been conducted evaluating the possible role of metformin as an anti-cancer drug.

In head and neck melanoma (HNM), the impact of DM2 and metformin on the outcome of cancer patients remains an unanswered question. Therefore, the main objective of this population-based multicenter cohort study was to evaluate the prognostic roles of DM2 and metformin in primarily resected HNM patients.

## 2. Materials and Methods

### 2.1. Patient Selection and Data Collection

This multicenter retrospective cohort study analyzed an Eastern Bavarian cohort of adult patients with a primarily diagnosed and in sano resected HNM, as previously described [16]. The present retrospective cohort study thereby used a population-based approach, including a representative population of around 2.3 million residents. Hereby, a comprehensive collection of data was accessed by using resources of the Tumor Center Regensburg. Primary diagnosis as well as tumor resection were carried out between 1 January 2010 and 31 December 2017; patients with a history of previous HNM, non in sano resection, incomplete tumor characteristics, and deficient individual tumor staging were excluded. Additionally, patients having received neoadjuvant treatment modalities were taken out of the current set of data. Staging was based on the guidelines of the American Joint Committee on Cancer (AJCC) cancer staging and manual in its 7th edition [17]. The clinical as well as pathohistological patient data were analyzed by accessing written and electronical medical records. The Charlson comorbidity index (CCI) was calculated as previously described and without taking HNM into account [18]. Routine medication was retrieved from individual medication plans and previous physician's letters prior to the diagnosis of HNM. Adjuvant treatment was based on the recommendation of the multidisciplinary tumor board, and chemo- and/or immunotherapy was used accordingly. The definition of disease relapse comprised local disease recurrence or diagnosis of distant metastasis by radiologic evidence, with clinical correlation or histologic confirmation with biopsy. The survival data were calculated based on follow-up data from medical records, death certificates, registration offices, as well as registry data. In this regard, recurrence-free survival (RFS) was assessed. The mean follow-up time for the entire cohort of HNM patients was 4.5 years (median 4.7 years).

### 2.2. Statistical Analysis

In descriptive statistics, categorical variables were analyzed using a Pearson's chi-squared test. Regarding the metric variables, differences in the means were evaluated based on Student's *t*-test in case of log-normal distribution, otherwise a Mann–Whitney

U-test was applied. For the survival analysis, uni- as well as multivariable statistics were applied. In this regard, the timespan between date of tumor resection and date of death, date of renewed tumor manifestation, or date last alive until the cut-off date of 31 March 2020 was chosen. The survival curves for recurrence-free survival (RFS) were created using the Kaplan–Meier method. Regarding the RFS, distinct curves for local, loco-regional, and distant RFS were calculated. The log-rank test was used to determine differences in survival. For the multivariable survival analysis, a risk adjustment for sex, age, AJCC stage, presence of ulceration, DM2, as well as for the intake of routine medication (ASA, statins, metformin) was achieved. The results include hazard ratios (HRs) and 95% confidence intervals (CIs). A *p*-value < 0.05 was considered statistically significant. All analyses were performed using IBM SPSS Statistics Version 26.0 (IBM Corp., Armonk, NY, USA).

## 3. Results

### 3.1. Characterization of the Patient Cohort

In the present retrospective population-based HNM cohort study, 222 (58.1%) patients were male (Table 1). For all HNM patients, the median age was 69.2 years, whereas the diabetic cases displayed a median age of 74.1 years. In this regard, a diagnosis of DM2 was highly correlated with increased age (*p* < 0.001), comorbidities according the Charlson comorbidity index beyond DM2 (*p* = 0.010), and an increased body mass index (BMI; *p* = 0.000). Hereby, the diabetic HNM patients received more likely routine medication such as statins or acetylsalicylic acid (ASA; *p* < 0.001 and *p* = 0.001, respectively; Table 1). Most of the diabetic HNM patients (57.4%) received metformin as a routine antidiabetic treatment (Table 1). In line with the diagnosis of DM2, metformin intake significantly correlated with increased age and obesity (*p* = 0.017 and *p* < 0.001, respectively; Table 2). Additionally, more female HNM patients received metformin as an antidiabetic drug (*p* = 0.009; Figure 1).

**Table 1.** Patients clinico-pathological characteristics according to a diagnosis of diabetes (*n* = 382).

| Variable | | No | | Yes | | Total | | χ2 |
|---|---|---|---|---|---|---|---|---|
| | | *N* | Column % | *N* | Column % | *N* | Column % | *p* |
| Sex | Male | 188 | 59.9% | 34 | 50.0% | 222 | 58.1% | 0.135 |
| | Female | 126 | 40.1% | 34 | 50.0% | 160 | 41.9% | |
| Age at diagnosis | <50 | 66 | 21.0% | 1 | 1.5% | 67 | 17.5% | <0.001 |
| | 50–59 | 52 | 16.6% | 3 | 4.4% | 55 | 14.4% | |
| | 60–69 | 59 | 18.8% | 17 | 25.0% | 76 | 19.9% | |
| | 70–79 | 91 | 29.0% | 32 | 47.1% | 123 | 32.2% | |
| | ≥80.0 | 46 | 14.6% | 15 | 22.1% | 61 | 16.0% | |
| Charlson comorbidity index (without diabetes) | 0 | 260 | 82.8% | 47 | 69.1% | 307 | 80.4% | 0.013 |
| | 1 | 39 | 12.4% | 14 | 20.6% | 53 | 13.9% | |
| | 2 | 10 | 3.2% | 2 | 2.9% | 12 | 3.1% | |
| | 3 | 4 | 1.3% | 4 | 5.9% | 8 | 2.1% | |
| | 4 | 0 | 0.0% | 1 | 1.5% | 1 | 0.3% | |
| | 5 | 1 | 0.3% | 0 | 0.0% | 1 | 0.3% | |
| Charlson comorbidity index (without diabetes) | 0 | 260 | 82.8% | 47 | 69.1% | 307 | 80.4% | 0.010 |
| | ≥1 | 54 | 17.2% | 21 | 30.9% | 75 | 19.6% | |
| BMI | Underweight | 1 | 0.3% | 0 | 0.0% | 1 | 0.3% | <0.001 |
| | Normal weight | 95 | 30.3% | 4 | 5.9% | 99 | 25.9% | |
| | Overweight | 113 | 36.0% | 25 | 36.8% | 138 | 36.1% | |
| | Obesity | 62 | 19.7% | 32 | 47.1% | 94 | 24.6% | |
| | N/A. | 43 | 13.7% | 7 | 10.3% | 50 | 13.1% | |
| BMI | Under-normal weight | 96 | 30.6% | 4 | 5.9% | 100 | 26.2% | <0.001 |
| | Overweight–obese | 175 | 55.7% | 57 | 83.8% | 232 | 60.7% | |
| | N/A | 43 | 13.7% | 7 | 10.3% | 50 | 13.1% | |
| Metformin | No | 314 | 100.0% | 29 | 42.6% | 343 | 89.8% | <0.001 |
| | Yes | 0 | 0.0% | 39 | 57.4% | 39 | 10.2% | |
| Statin | No | 269 | 85.7% | 42 | 61.8% | 311 | 81.4% | <0.001 |
| | Yes | 45 | 14.3% | 26 | 38.2% | 71 | 18.6% | |
| ASA | No | 256 | 81.5% | 43 | 63.2% | 299 | 78.3% | 0.001 |
| | Yes | 58 | 18.5% | 25 | 36.8% | 83 | 21.7% | |

**Table 1.** *Cont.*

| Variable | | No | | Yes | | Total | | χ2 |
|---|---|---|---|---|---|---|---|---|
| | | *N* | Column % | *N* | Column % | *N* | Column % | *p* |
| Localization ICDO-3 | C44.0 Lip skin | 1 | 0.3% | 1 | 1.5% | 2 | 0.5% | 0.216 |
| | C44.1 Eyelid | 15 | 4.8% | 3 | 4.4% | 18 | 4.7% | |
| | C44.2 Outer ear | 53 | 16.9% | 6 | 8.8% | 59 | 15.4% | |
| | C44.3 Other parts of the face | 143 | 45.5% | 39 | 57.4% | 182 | 47.6% | |
| | C44.4 Scalp and neck | 102 | 32.5% | 19 | 27.9% | 121 | 31.7% | |
| Localization | Scalp and neck | 102 | 32.5% | 19 | 27.9% | 121 | 31.7% | 0.465 |
| | Face | 212 | 67.5% | 49 | 72.1% | 261 | 68.3% | |
| AJCC stage | IA | 172 | 54.8% | 33 | 48.5% | 205 | 53.7% | 0.595 |
| | IB | 53 | 16.9% | 16 | 23.5% | 69 | 18.1% | |
| | II | 72 | 22.9% | 16 | 23.5% | 88 | 23.0% | |
| | III | 17 | 5.4% | 3 | 4.4% | 20 | 5.2% | |
| AJCC stage | I | 225 | 71.7% | 49 | 72.1% | 274 | 71.7% | 0.947 |
| | II + III | 89 | 28.3% | 19 | 27.9% | 108 | 28.3% | |
| Tumor thickness (mm) | <1 | 190 | 60.5% | 38 | 55.9% | 228 | 59.7% | |
| | 1–2 | 50 | 15.9% | 13 | 19.1% | 63 | 16.5% | |
| | 2–4 | 35 | 11.1% | 7 | 10.3% | 42 | 11.0% | |
| | >4 | 39 | 12.4% | 10 | 14.7% | 49 | 12.8% | |
| Tumor size (pT) | 1 | 195 | 62.1% | 39 | 57.4% | 234 | 61.3% | 0.908 |
| | 2 | 53 | 16.9% | 13 | 19.1% | 66 | 17.3% | |
| | 3 | 30 | 9.6% | 7 | 10.3% | 37 | 9.7% | |
| | 4 | 36 | 11.5% | 9 | 13.2% | 45 | 11.8% | |
| Nodal status (pN) | 0 | 297 | 94.6% | 65 | 95.6% | 362 | 94.8% | 0.737 |
| | 1 | 17 | 5.4% | 3 | 4.4% | 20 | 5.2% | |
| Ulceration | no | 249 | 79.3% | 54 | 79.4% | 303 | 79.3% | 0.715 |
| | yes | 62 | 19.7% | 14 | 20.6% | 76 | 19.9% | |
| | N/A | 3 | 1.0% | 0 | 0.0% | 3 | 0.8% | |
| Histological subgroup | Lentigo-maligna melanoma | 125 | 39.8% | 29 | 42.6% | 154 | 40.3% | 0.678 |
| | Nodular melanoma | 46 | 14.6% | 13 | 19.1% | 59 | 15.4% | |
| | Superficial spreading melanoma | 103 | 32.8% | 19 | 27.9% | 122 | 31.9% | |
| | Melanoma n. o. s. and other | 40 | 12.7% | 7 | 10.3% | 47 | 12.3% | |
| Surgical margin (mm) | <5 | 75 | 23.9% | 18 | 26.5% | 93 | 24.3% | 0.018 |
| | 5–9 | 122 | 38.9% | 22 | 32.4% | 144 | 37.7% | |
| | ≥10 | 117 | 37.3% | 26 | 38.2% | 143 | 37.4% | |
| | N/A | 0 | 0.0% | 2 | 2.9% | 2 | 0.5% | |
| Neck dissection | No | 294 | 93.6% | 58 | 85.3% | 352 | 92.1% | 0.021 |
| | Yes | 20 | 6.4% | 10 | 14.7% | 30 | 7.9% | |
| Adjuvant radiotherapy | No | 310 | 98.7% | 68 | 100.0% | 378 | 99.0% | 0.349 |
| | Yes | 4 | 1.3% | 0 | 0.0% | 4 | 1.0% | |
| Adjuvant chemo/immunotherapy | No | 299 | 95.2% | 65 | 95.6% | 364 | 95.3% | 0.897 |
| | Yes | 15 | 4.8% | 3 | 4.4% | 18 | 4.7% | |
| | Total | 314 | 100.0% | 68 | 100.0% | 382 | 100.0% | |

*N*: number of subjects; %: percentage of subjects.

**Table 2.** Patients clinico-pathological characteristics according to metformin use (*n* = 382).

| Variable | | No | | Yes | | Total | | χ2 |
|---|---|---|---|---|---|---|---|---|
| | | *N* | Column % | *N* | Column % | *N* | Column % | *p* |
| Sex | Male | 207 | 60.3% | 15 | 38.5% | 222 | 58.1% | 0.009 |
| | Female | 136 | 39.7% | 24 | 61.5% | 160 | 41.9% | |
| Age at diagnosis | <50 | 66 | 19.2% | 1 | 2.6% | 67 | 17.5% | 0.017 |
| | 50–59 | 53 | 15.5% | 2 | 5.1% | 55 | 14.4% | |
| | 60–69 | 66 | 19.2% | 10 | 25.6% | 76 | 19.9% | |
| | 70–79 | 106 | 30.9% | 17 | 43.6% | 123 | 32.2% | |
| | ≥80 | 52 | 15.2% | 9 | 23.1% | 61 | 16.0% | |
| Charlson comorbidity index (without diabetes) | 0 | 277 | 80.8% | 30 | 76.9% | 307 | 80.4% | 0.537 |
| | 1 | 46 | 13.4% | 7 | 17.9% | 53 | 13.9% | |

**Table 2.** *Cont.*

| | | Metformin | | | | | | χ2 |
|---|---|---|---|---|---|---|---|---|
| | | No | | Yes | | Total | | |
| Variable | | N | Column % | N | Column % | N | Column % | p |
| | 2 | 12 | 3.5% | 0 | 0.0% | 12 | 3.1% | |
| | 3 | 6 | 1.7% | 2 | 5.1% | 8 | 2.1% | |
| | 4 | 1 | 0.3% | 0 | 0.0% | 1 | 0.3% | |
| | 5 | 1 | 0.3% | 0 | 0.0% | 1 | 0.3% | |
| Charlson comorbidity index (without diabetes) | 0 | 277 | 80.8% | 30 | 76.9% | 307 | 80.4% | 0.568 |
| | ≥1 | 66 | 19.2% | 9 | 23.1% | 75 | 19.6% | |
| BMI | Underweight | 1 | 0.3% | 0 | 0.0% | 1 | 0.3% | <0.001 |
| | Normal weight | 96 | 28.0% | 3 | 7.7% | 99 | 25.9% | |
| | Overweight | 124 | 36.2% | 14 | 35.9% | 138 | 36.1% | |
| | Obesity | 74 | 21.6% | 20 | 51.3% | 94 | 24.6% | |
| | N/A | 48 | 14.0% | 2 | 5.1% | 50 | 13.1% | |
| BMI | Under-normal weight | 97 | 28.3% | 3 | 7.7% | 100 | 26.2% | 0.002 |
| | Overweight–obese | 198 | 57.7% | 34 | 87.2% | 232 | 60.7% | |
| | N/A | 48 | 14.0% | 2 | 5.1% | 50 | 13.1% | |
| Statin | No | 282 | 82.2% | 29 | 74.4% | 311 | 81.4% | 0.232 |
| | Yes | 61 | 17.8% | 10 | 25.6% | 71 | 18.6% | |
| ASA | No | 274 | 79.9% | 25 | 64.1% | 299 | 78.3% | 0.024 |
| | Yes | 69 | 20.1% | 14 | 35.9% | 83 | 21.7% | |
| Localization ICDO-3 | C44.0 Lip skin | 1 | 0.3% | 1 | 2.6% | 2 | 0.5% | 0.445 |
| | C44.1 Eyelid | 16 | 4.7% | 2 | 5.1% | 18 | 4.7% | |
| | C44.2 Outer ear | 54 | 15.7% | 5 | 12.8% | 59 | 15.4% | |
| | C44.3 Other parts of the face | 164 | 47.8% | 18 | 46.2% | 182 | 47.6% | |
| | C44.4 Scalp and neck | 108 | 31.5% | 13 | 33.3% | 121 | 31.7% | |
| Localization | Scalp and neck | 108 | 31.5% | 13 | 33.3% | 121 | 31.7% | 0.814 |
| | Face | 235 | 68.5% | 26 | 66.7% | 261 | 68.3% | |
| AJCC stage | IA | 185 | 53.9% | 20 | 51.3% | 205 | 53.7% | 0.590 |
| | IB | 59 | 17.2% | 10 | 25.6% | 69 | 18.1% | |
| | II | 81 | 23.6% | 7 | 17.9% | 88 | 23.0% | |
| | III | 18 | 5.2% | 2 | 5.1% | 20 | 5.2% | |
| AJCC stage | I | 244 | 71.1% | 30 | 76.9% | 274 | 71.7% | 0.447 |
| | II + III | 99 | 28.9% | 9 | 23.1% | 108 | 28.3% | |
| Tumor thickness (mm) | <1 | 204 | 59.5% | 24 | 61.5% | 228 | 59.7% | 0.938 |
| | 1–2 | 57 | 16.6% | 6 | 15.4% | 63 | 16.5% | |
| | 2–4 | 37 | 10.8% | 5 | 12.8% | 42 | 11.0% | |
| | >4 | 45 | 13.1% | 4 | 10.3% | 49 | 12.8% | |
| Tumor size (TNM) | 1 | 209 | 60.9% | 25 | 64.1% | 234 | 61.3% | 0.752 |
| | 2 | 60 | 17.5% | 6 | 15.4% | 66 | 17.3% | |
| | 3 | 32 | 9.3% | 5 | 12.8% | 37 | 9.7% | |
| | 4 | 42 | 12.2% | 3 | 7.7% | 45 | 11.8% | |
| Nodal status (TNM) | 0 | 325 | 94.8% | 37 | 94.9% | 362 | 94.8% | 0.975 |
| | 1 | 18 | 5.2% | 2 | 5.1% | 20 | 5.2% | |
| Ulceration | No | 270 | 78.7% | 33 | 84.6% | 303 | 79.3% | 0.625 |
| | Yes | 70 | 20.4% | 6 | 15.4% | 76 | 19.9% | |
| | N/A | 3 | 0.9% | 0 | 0.0% | 3 | 0.8% | |
| Histological subgroup | Lentigo-maligna melanoma | 140 | 40.8% | 14 | 35.9% | 154 | 40.3% | 0.555 |
| | Nodular melanoma | 53 | 15.5% | 6 | 15.4% | 59 | 15.4% | |
| | Superficial spreading melanoma | 106 | 30.9% | 16 | 41.0% | 122 | 31.9% | |
| | Melanoma n. o. s. and other | 44 | 12.8% | 3 | 7.7% | 47 | 12.3% | |
| Surgical margin (mm) | <5 | 86 | 25.1% | 7 | 17.9% | 93 | 24.3% | 0.152 |
| | 5–9 | 131 | 38.2% | 13 | 33.3% | 144 | 37.7% | |
| | ≥10 | 125 | 36.4% | 18 | 46.2% | 143 | 37.4% | |
| | N/A | 1 | 0.3% | 1 | 2.6% | 2 | 0.5% | |
| Neck dissection | No | 316 | 92.1% | 36 | 92.3% | 352 | 92.1% | 0.969 |
| | Yes | 27 | 7.9% | 3 | 7.7% | 30 | 7.9% | |
| Adjuvant radiotherapy | No | 339 | 98.8% | 39 | 100.0% | 378 | 99.0% | 0.498 |
| | Yes | 4 | 1.2% | 0 | 0.0% | 4 | 1.0% | |
| Adjuvant chemo/immunotherapy | No | 326 | 95.0% | 38 | 97.4% | 364 | 95.3% | 0.504 |
| | Yes | 17 | 5.0% | 1 | 2.6% | 18 | 4.7% | |
| | Total | 343 | 100.0% | 39 | 100.0% | 382 | 100.0% | |

*N*: number of subjects; %: percentage of subjects.

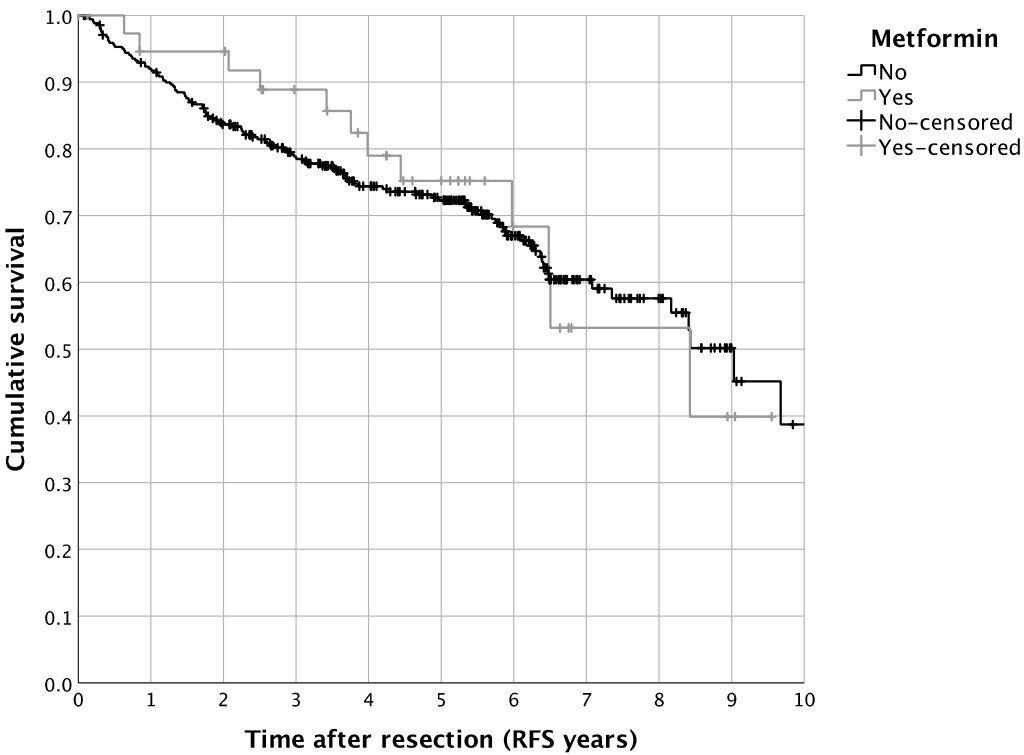

**Figure 1.** Survival of HNM patients differentiated by metformin use: Kaplan–Meier curves for RFS (*p* = 0.806, *n* = 382, patients with routine metformin medication: *n* = 39).

### 3.2. Impact of DM2 and Metformin Intake on Patient Survival

The univariate survival analysis revealed a 5-year RFS of 72.5%, with a median 5-year RFS of 8.4 years. The diabetic patients showed a reduced 5-year RFS in contrast to the non-diabetic cases (64.0% vs. 74.5%, *p* = 0.016; Figure 2). In contrast, a routine metformin treatment did not alter the RFS in the complete cohort of HNM patients (*p* = 0.805; Figure 1). Besides the univariate survival analysis, we performed a multivariate Cox regression to adjust for covariables such as age, BMI, statin use, intake of ASA, tumor size, nodal status, ulceration, and distinct tumor histologic types. For the entire cohort of HNM patients, diabetic cases showed, in line with the results of the univariate statistics, reduced RFS (HR = 1.980, 95% CI = 1.108–3.538, *p* = 0.021; Table 3). For metformin use, the multivariate Cox regression analysis resulted in a significantly more favorable outcome in comparison to the controls (HR = 0.396, 95% CI = 0.177–0.884, *p* = 0.024; Table 3). Additionally, we analyzed distinct recurrence rates in detail, distinguishing a cumulative recurrence rate, a cumulative locoregional recurrence rate, as well as a cumulative distant recurrence rate (Table 3). Especially when focusing on the cumulative locoregional recurrence rate, both effects, the detrimental one of DM2 (HR = 4.173, 95% CI = 1.628–10.697, *p* = 0.003) as well as the beneficial one of a routine metformin use (HR = 0.135, 95% CI = 0.031–0.589, *p* = 0.008) were even more pronounced (Table 3). Besides the survival analysis of the complete cohort of HNM patients, we performed a more detailed perspective analysis on the cohort of diabetic patients. Hereby, univariate Cox regression displayed a significantly beneficial effect of metformin use on the RFS of diabetic HNM patients (HR = 0.417, 95% CI = 0.201–0.868, *p* = 0.019; Table 4). After applying multivariate statistics, this beneficial effect on the RFS of HNM patients was validated in the subgroup analysis (HR = 0.352, 95% CI = 0.135–0.913, *p* = 0.032; Table 4).

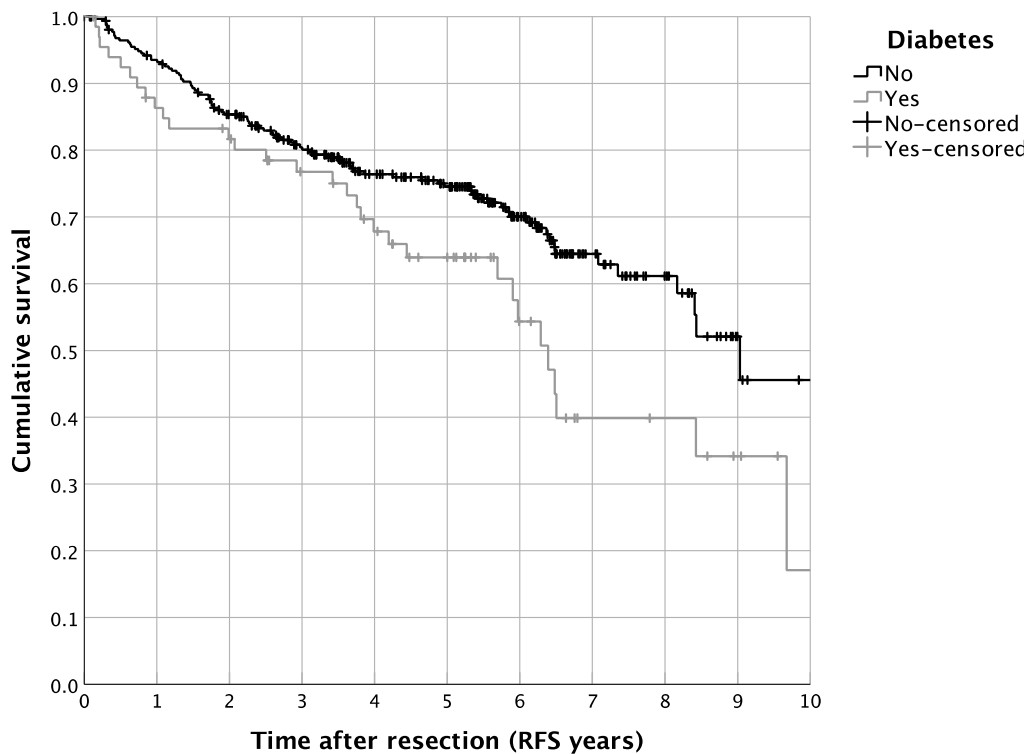

**Figure 2.** Survival of HNM patients differentiated by a diagnosis of diabetes: Kaplan–Meier curves for RFS ($p$ = 0.018, $n$ = 382, diabetic patients: $n$ = 68).

**Table 3.** Synopsis of the results from the multivariable Cox regression for RFS and cumulative recurrence rates according to metformin use and diabetes in the complete cohort.

| Complete Cohort | **Multivariable Cox Regression \*** | | | |
|---|---|---|---|---|
| | $p$ | **Hazard Ratio** | **Lower 95%-CI** | **Upper 95%-CI** |
| Recurrence-free survival | | | | |
| Metformin yes vs. no | 0.024 | 0.396 | 0.177 | 0.884 |
| Diabetes yes vs. no | 0.021 | 1.980 | 1.108 | 3.538 |
| Cumulative recurrence rate | | | | |
| Metformin yes vs. no | 0.061 | 0.325 | 0.100 | 1.054 |
| Diabetes yes vs. no | 0.141 | 1.915 | 0.807 | 4.547 |
| Cumulative locoregional recurrence rate | | | | |
| Metformin yes vs. no | 0.008 | 0.135 | **0.031** | 0.589 |
| Diabetes yes vs. no | 0.003 | 4.173 | 1.628 | 10.697 |
| Cumulative distant recurrence rate | | | | |
| Metformin yes vs. no | 0.702 | 1.356 | 0.286 | 6.426 |
| Diabetes yes vs. no | 0.124 | 0.370 | 0.104 | 1.312 |

\* adjusted for age at diagnosis, body mass index, statin use, ASA, tumor size, nodal status, ulceration, and histologic type (both metformin and diabetes included in the model).

**Table 4.** Synopsis of the results from the multivariable Cox regression for RFS and cumulative recurrence rates according to metformin use in the diabetes complete cohort.

| Diabetes Cohort | Multivariable Cox Regression * | | | |
|---|---|---|---|---|
| | *p* | Hazard Ratio | Lower 95%-CI | Upper 95%-CI |
| Recurrence-free survival | | | | |
| Metformin yes vs. no | 0.032 | 0.352 | 0.135 | 0.913 |
| Cumulative recurrence rate | | | | |
| Metformin yes vs. no | 0.869 | 0.855 | 0.133 | 5.479 |
| Cumulative locoregional recurrence rate | | | | |
| Metformin yes vs. no | 0.964 | 0.952 | 0.109 | 8.305 |
| Cumulative distant recurrence rate | | | | |
| Metformin yes vs. no | - [&] | - | - | - |

* adjusted for age at diagnosis, body mass index, statin use, ASA, tumor size, nodal status, ulceration, and histologic type (both metformin and diabetes included in the model); [&] due to the small number of events, the model did not converge.

## 4. Discussion

Since early findings about its possible antineoplastic characteristics were published [15], extensive clinical as well as basic research has been conducted to characterize the role of metformin in cancer. One of several milestones in this regard is certainly the work of Coyle et al., analyzing several observational studies, including over 24,000 participants and spreading over a large variety of cancer entities [19]. Although it pointed out the antitumorigenic effects of metformin intake on prostate and colorectal cancer, no information was provided for the possible anti-cancer effects of metformin in melanoma [19]. Over a decade later, not only retrospective studies about the role of metformin in cancer patients were conducted, but also prospective clinical trials were initiated, especially in melanoma patients: a selected collective of metastatic melanoma patients who did not respond to chemotherapy received therapeutic doses of metformin for up to six months [20]. Although promising in vitro results about the possible role of metformin as an anti-cancer drug in melanoma inspired this clinical trial [21], the outcome was sobering [20]: out of 17 enrolled cases, none of the patients showed a complete or at least partial response after 6 months of metformin treatment [20]. However, these disappointing results need to be more closely analyzed: every patient in that study had already received classic chemotherapeutic/immunotherapeutic treatments (chemotherapy, BRAF inhibitor, or ipilimumab) or was not eligible for immunotherapy [20]. In this specific collective of advanced metastatic melanoma patients, a last-line treatment based on metformin monotherapy might be a quite ambitious approach. In contrast to this, a prospective study in early breast cancer demonstrated that a neoadjuvant therapy based on metformin treatment for a median of 18 days resulted in promising cellular as well as clinical anti-cancer effects [22].

In the retrospective setting of our multicenter cohort study, metformin intake was due to a clinical diagnosis of DM2 and, therefore, metformin primarily did not act as an anti-cancer drug at first sight. Recent data about diabetes prevalence in Germany showed comparable but even lower rates than presented in our study cohort [23]. However, when analyzing the role of metformin on the outcome of melanoma patients in detail, the multivariate survival analysis clearly indicated a benefit in the RFS in diabetic melanoma patients in the present retrospective cohort study. Obviously, descriptive observations based on retrospective data cannot proof an anti-cancer effect of metformin in melanoma. Hereby, in vitro results from different groups might be helpful to understand the underlying mechanisms: on one hand, metformin was characterized to have several indirect effects on cancer cells, mainly mediated by lowering the circulating insulin levels [19,21]. On the other hand, most anti-tumor effects of metformin are linked to the direct inhibition of the mammalian target of rapamycin complex 1 (mTORC1) [21]. In simple terms, the

mTOR complex is involved in several cellular processes such as protein synthesis and was shown to be upregulated in certain malignancies. Metformin, as an inhibitor of the mTOR pathway, therefore directly inhibits tumor cell proliferation and induces apoptosis and cell cycle arrest in vitro [24,25].

Besides the in vitro studies regarding the role of metformin in cancer, the clinical application of metformin is directly due to an underlying DM2. In this regard, the impact of DM2, especially in developed countries, is unquestionable: in a comprehensive registry-based cohort study in the United Kingdom, Heald et al. attributed to diabetic patients in comparison to non-diabetic equivalents a loss of 1.7 years of life [26]. Unquestionably, DM2 is a major public health problem and can certainly be classified, even in times of COVID-19, as a disease reaching epidemic proportions [27]. Especially in malignant disease, the role of DM2 has recently been characterized: DM2 not only increases the patients' risk to develop various malignant diseases, but additionally increases the peri- as well as the postoperative morbidity and mortality of patients undergoing extensive resections of malignant lesions [28]. The underlying mechanisms are manifold: besides hyperglycemia and elevated blood insulin levels with the stimulation of the IGF-1 pathway, several adipokines, secreted by the adipose tissue, promote carcinogenesis [28,29]. It is therefore not surprising that numerous studies attributed to diabetic cancer patients a fundamentally reduced outcome [30,31]. For HNM, however, the literature about the role of DM2 in modulating carcinogenesis and tumor progression is quite thin: in a meta-analysis comprising nine independent cohorts, the authors attributed to patients with DM2 a slightly increased risk to develop malignant melanoma; however, whether this also translated into distinct survival rates between diabetic and non-diabetic melanoma-patients remained unclear [32]. In the present retrospective population-based cohort study on HNM uni- as well as multivariate survival analyses revealed a significantly impaired outcome for DM2 patients. To our knowledge, this is the first cohort study on HNM, attributing to DM2 patients a clearly reduced RFS. Even after adjusting for possible confounders, the negative impact of DM2 in HNM patients persisted. However, as a non-neglectable limitation of the present cohort study, the retrospective approach entails some pitfalls. Data on the diagnosis of DM2, for instance, and the treatment with metformin were collected with the aid of written and electronical medical records, which might in fact not reflect the status quo of each patient.

## 5. Conclusions

In conclusion, our results emphasize the relevance of DM2 as a relevant predictor of outcome in HNM patients. Metformin as the most prevalent antidiabetic drug for DM2 was shown to significantly ameliorate the survival of patients suffering from HNM. In this regard, appropriate glucose-lowering therapies might provide beneficial outcomes especially in patients with this selected cancer entity. Related to this, antidiabetic drugs such as metformin might be promising candidates for future treatment regimes for malignant diseases such as HNM.

**Author Contributions:** Conceptualization: S.S. (Steffen Spoerl) and G.S.; data curation: S.S. (Steffen Spoerl), M.G., S.S. (Susanne Schimnitz), J.T. and R.F.; formal analysis: S.S. (Steffen Spoerl) M.G., S.S. (Susanne Schimnitz) and G.S.; funding acquisition: S.S. (Steffen Spoerl) and G.S.; investigation: S.S. (Steffen Spoerl), R.F. and G.S.; methodology: S.S. (Steffen Spoerl), M.G., J.T., N.L., S.R.L., S.S. (Silvia Spoerl) and G.S.; project administration: S.S. (Steffen Spoerl) and G.S.; resources: T.E.R., M.G. and G.S.; software: S.S. (Steffen Spoerl) and M.G.; supervision: G.S. and T.E.R.; validation: S.S. (Steffen Spoerl) and G.S.; visualization: M.G., J.T. and G.S.; writing—original draft: S.S. (Steffen Spoerl) and G.S.; writing—review & editing: S.S. (Steffen Spoerl), M.G., S.S. (Susanne Schimnitz), J.T., R.F., S.R.L., T.E., N.L., S.S. (Silvia Spoerl), T.E.R. and G.S. All authors have read and agreed to the published version of the manuscript.

**Funding:** Open Access funding was supported and enabled by the Else Kröner-Fresenius-Stiftung, Germany.

**Institutional Review Board Statement:** The protocol was approved by the University of Regensburg Ethics Committee (GeschZ 19–1530-104), and the study was conducted in accordance with the ethical standards of the declaration of Helsinki.

**Informed Consent Statement:** Based on a retrospective analysis and a fully anonymized set of clinical data and in agreement with the decision of the Regensburg University Ethics Committee, signing an informed consent form was not required.

**Data Availability Statement:** Data can be obtained by scientists that work independently from the industry on request. Data are not stored on publicly available servers.

**Acknowledgments:** We thank the Else Kröner-Fresenius-Stiftung, Germany for supporting the present research project. Additionally, we thank Susanne Schimnitz for her efforts.

**Conflicts of Interest:** The authors declare no conflict of interest.

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
