# Peer review of "Prognostic Relevance of Type 2 Diabetes and Metformin Treatment in Head and Neck Melanoma: Results from a Population-Based Cohort Study"

_curroncol, doi:10.3390/curroncol29120758_

Round 1

Reviewer 2 Report

This paper is very interesting and represents the statistical results of one cohort of patients with head and neck melanoma. Although, of the 382 participating patients, only 68 have been diagnosed with diabetes (see Table 1). Al igual que la estadística de los casos tratados con metformina es algo reducida (15 and 24 for male and female, see Table 2).

Even so, la statistic confirms the beneficial effect of metformin use on the RFS of diabetic HNM patients.

I suggest clarifying, as a limitation the small number of patients with diabetes in a cohort of cancer patients.

Reviewer 3 Report

This study aimed to explore the role of DM2 and antidiabetic drugs on outcome of head and neck melanoma (HNM) patients in a population based study. They identified that DM2 is significantly correlated to the poor survival outcome of these patients. Metformin intake could improve their survival outcome. I think this manuscript could be accepted for publication after necessary revisions.

1. The author used 7th edition of the TNM staging system in the study. However, the 8th edition of the TNM staging system is widly used in cancer diagnosis and management in recent years. So, the stage of each patient should be transformed into new staging according to their T, N, and M stage.

2. The prognosis of cancer patients was influenced by many factors. How about the impact of laboratory examination results on the prognosis of these patients? 

3. The figures and tables could be improved.

Round 2

Reviewer 1 Report

After corrections (minimal, but sufficient) the paper is more readable and suitable for publication. For future work I suggest the authors to follow books cited in the previous review. This will improve their work.

Reviewer 3 Report

This manuscript could be accepted for publication